# Mastering Pixel-Based Reinforcement Learning via Positive Unlabeled Policy-Guided Contrast

## Abstract

Real-world reinforcement learning has received a significant amount of attention very recently. A fundamental yet challenging problem in this learning paradigm is perceiving real-world environmental information, such that *pixel-based* reinforcement learning emerges, which aims to learn representation from visual observations for policy optimization. In this article, we profoundly elaborate the frameworks of benchmark methods and demonstrate a long-standing *paradox* challenging current methods: in different training phases, exploring visual semantic information can improve and prevent the performance of the learned feature representations from improving. In practice, we further disclose that the over-redundancy issue generally halts the rise of sample efficiency among baseline methods. To remedy the uncovered deficiency of existing methods, we introduce a novel plug-and-play method for pixel-based reinforcement learning. Our model involves the *positive unlabeled policy-guided contrast* to learn jointly anti-redundant and policy-optimization-relevant visual semantic information during training. To sufficiently elucidate the proposed method's innate superiority, we revisit the pixel-based reinforcement learning paradigm from the information theory perspective. The theoretical evidence proves that the proposed model can achieve the tighter lower bound of the mutual information between the policy optimization-related information and the information of the representation derived by the encoder. To carry out the evaluation of our model, we conduct extensive benchmark experiments and illustrate the superior performance of our method over existing methods with respect to the pixel observation environments.

## 1 Introduction

**Background**. In recent years, Reinforcement Learning (RL) has made significant strides in various real-world domains, from Game AI Silver et al. (2017; 2018); Vinyals et al. (2019) to Robotics Andrychowicz et al. (2020). Such successes are majorly built on well-designed state space, which is orthogonal to real-world applications. Thus, pixel-based RL emerges to directly learn the state from natural data, e.g., sound, images, and videos. However, such a learning paradigm suffers from sample inefficiency, especially in intricate tasks with multi-dimensional pixel observations (e.g., RGB images). Drawing inspiration from the success of data augmentation and self-supervised learning in image and video processing, pixel-based RL seeks to learn more visual semantic information from pixel observation to enhance the sample efficiency, mainly through data augmentation Yarats et al. (2021a; 2022) or self-supervised contrastive learning Laskin et al. (2020b); Choi et al. (2023).

**Challenges**. To profoundly explore the intrinsic challenges in the field of pixel-based RL, we demonstrate the foundational frameworks for illustrating the learning paradigm of benchmark methods, including DQN Hosu & Rebedea (2016), CURL Laskin et al. (2020b), DrQ-v2 Yarats et al. (2022), and our proposed method in Figure 1 a). For determining the defects related to the innate property of benchmark methods, we perform further experimental analysis concerning the reward curve, which is demonstrated in Figure 1 b). Specifically, DQN adopts the traditional reward-based training paradigm of RL, such that the long-standing issue arises from the limited sample efficiency degenerates the performance of DQN, which is proved by the empirical evidences that at the beginning of the curve in Figure 1 b), the reward derived by DQN is consistently lower than that of CURL

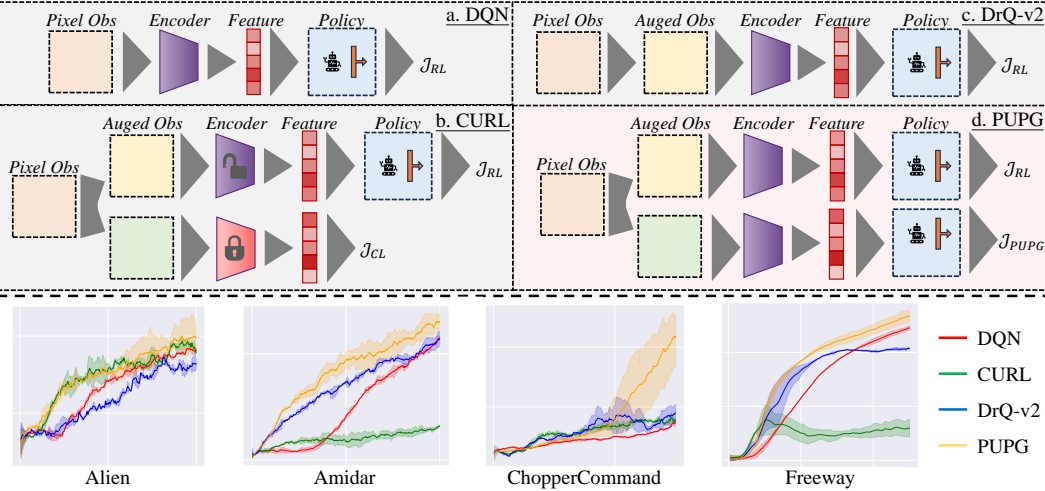

Figure 1: Simple diagram of DQN, CURL, DrQ-v2 and PUPG, and the reflection of local optimum on the reward curves.

and the proposed method. Furthermore, the encoder of DQN cannot learn the visual semantic information, and the sole reward-dependent training objective leads the encoder to learn limited semantic information from the pixel observations, resulting in the encoder training being easily trapped into a local optimum. Figure 1 b) demonstrates that the proposed method can consistently outperform DQN among benchmarks, substantiating the aforementioned statement. CURL leverages a contrast-based training paradigm to lead the encoder to learn the visual semantic information. This approach can relatively improve the sample efficiency for training the encoder. Nevertheless, the visual information derived by the encoder is superfluous to the policy network of RL since the contrast-based training paradigm is decoupled from the policy network, which interferes with the training of the RL model, resulting in degenerating the ultimate performance of the model. Additionally, CURL requires a wealth of negative samples for performing sufficient self-supervision, which dilutes model training efficiency. Figure 1 b) shows that CURL generally outperforms DQN at the beginning of training while eventually underperforming DQN, such that the aforementioned analyses of CURL learning paradigm are theoretically and empirically solid. DrQ-v2 introduces the data augmentation technique in the reward-based training paradigm of the encoder, which can improve the sample efficiency to a certain extent. However, the intrinsic defect of the reward-based training approach is not sufficiently addressed, i.e., the visual semantic information is not explored, so the training DrQ-v2 may still fall into a local optimum. Concretely, obtaining visual semantic information can improve the sample efficiency and the performance potential of the RL model. However, emphasizing the acquisition of visual semantic information causes interference upon the RL model's training, so the model still cannot reach the global optimum required for policy optimization. Furthermore, an essential defect challenges the benchmark methods, i.e., the visual representation learned by the encoder generally contains redundant information, which is another pivotal issue degenerating the sample efficiency and performance of the RL model.

**Contributions**. Exploring the visual semantic information is a double-edged sword, which jointly improves the sample efficiency and performance of the RL model while concentrating greater risk on the training interference and redundancy issues upon RL models. To this end, we propose *Positive Unlabeled Policy-Guided contrast for pixel-based reinforcement learning*, dubbed *PUPG*, to learn anti-redundant visual semantic information in a trade-off manner. The proposed method introduces a positive unlabeled contrastive learning paradigm to jointly avoid the over-dependence upon sufficient negative samples of the benchmark method and ratchet up the information entropy of the representation learned by the encoder during training. In this regard, PUPG can adequately mitigate the adverse effects towards the model training incurred by the limited sample efficiency of RL. The anti-redundancy property of the derived representation can fully guarantee the high optimization potential of the model. The guidance of policy optimization is further leveraged to control the learning of representations and wind up the acquisition of trivial visual semantic information, thereby preventing the model's training from getting trapped in undesired local optimum. The il-

lustrative example of the PUPG framework is shown in Figure 1 a). Without loss of generality, we revisit the pixel-based reinforcement learning paradigm from the information theory perspective and further demonstrate the intrinsic reasons behind the issues challenging benchmark methods and the theoretical merits of the proposed PUPG. Empirically, we conduct abundant experiments on various pixel-based RL environments, including Atari Towers et al. (2023) and DeepMind Control Suite (DMControl) Tunyasuvunakool et al. (2020), and the results prove that the proposed PUPG can consistently outperform the benchmark methods. The ablation study further demonstrates the effectiveness of each part of PUPG. The major **contributions** of this work are listed as follows:

- We disclose the long-standing issues challenging benchmark pixel-based RL methods: the duality of exploring the visual semantic information towards the training of the RL model and the over-redundancy problem generally existing in benchmarks.

- We propose the positive unlabeled policy-guided contrast for pixel-based RL, orthogonal to existing benchmark methods, which learns anti-redundant and policy-optimization-relevant visual semantic information for the training of the RL model.

- We theoretically rethink the learning paradigm of pixel-based RL from the information theory perspective and further establish that the proposed method can achieve the tighter lower bound of the mutual information between the policy optimization-related information and the information contained by the learned representation.

- We conduct extensive evaluations on various RL experimental settings, including Atari and DMControl, to empirically prove the effectiveness of each part of our proposed method.

## 2 RELATED WORKS

### 2.1 SELF-SUPERVISED CONTRASTIVE LEARNING IN COMPUTER VISION

In computer vision (CV), self-supervised contrastive learning aims to learn rich representations of high dimensional unlabeled data by bringing positive samples closer while separating negative samples from each otherBachman et al. (2019); Caron et al. (2020); Chen et al. (2020); Chen & He (2021); He et al. (2020); Zbontar et al. (2021); Grill et al. (2020). Tian et al. (2020) proposes a probabilistic contrastive loss, InfoNCE loss, to induce representations by leveraging positive and negative samples. Wu et al. (2018) implements InfoNCE with a memory bank. Chen et al. (2020) presents a framework for self-supervised contrastive learning without a memory bank but with a large batch size. He et al. (2020) use a dynamic dictionary with a queue to avoid using large batch size, and they also use the moving averaged encoder for the target data. Grill et al. (2020) use the momentum encoder to produce representations of the targets to stabilize the bootstrap step, enabling representations learning without negative samples. Zbontar et al. (2021) measure the cross-correlation matrix between the outputs to avoid collapse, which only needs positive samples.

### 2.2 PIXEL-BASED REINFORCEMENT LEARNING

Learning semantic information from pixel observations is a fundamental problem of RL in many environments that provide only pixel observations. Hosu & Rebedea (2016); van Hasselt et al. (2016); Hessel et al. (2018) regard pixel-based RL as an end-to-end optimization problem. Meanwhile, Stooke et al. (2021) tries to decouple representation learning from reinforcement learning. Moreover, Yarats et al. (2021b); Hafner et al. (2019); Lee et al. (2020) combines self-supervised learning and RL by implementing an auto-encoder as an auxiliary task. Works like Yarats et al. (2021a; 2022); Laskin et al. (2020a) only apply data augmentation to improve sample efficiency, achieving better performance than prior model-based methods. Contrastive learning is also considered in other works, Laskin et al. (2020b)learns the representations from visual inputs using the InfoNCE loss, Schwarzer et al. (2020) implements a negative-free self-supervised contrastive learning approach to imposing the similarity constraint between self-predictive and target representations. Li et al. (2022) analyzes the effect of various self-supervised learning tasks on pixel-based RL.

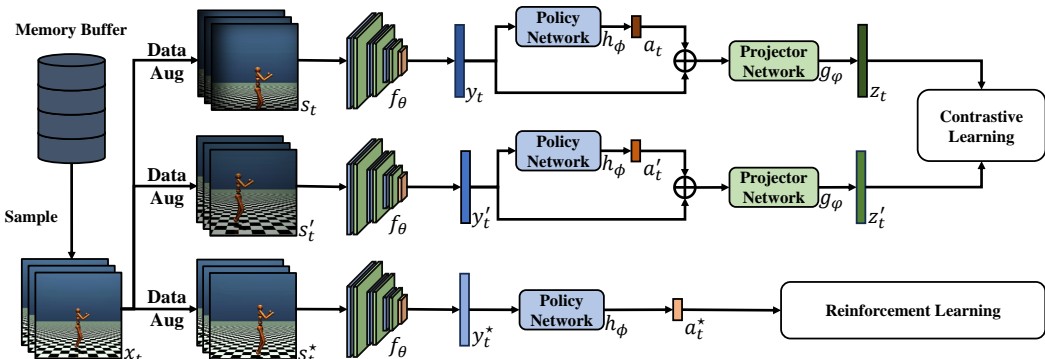

Figure 2: The framework of PUPG.

## 3 METHOD

### 3.1 PRELIMINARIES

Pixel-based RL is formulated as an infinite-horizon Markov Decision Process (MDP) Bellman (1957). Generally, such MDP can be described as a tuple $(\mathcal{X}, \mathcal{A}, \mathcal{P}, \mathcal{R}, \gamma, d_0)$, where $\mathcal{X}$ is the state space of images, $\mathcal{A}$ is the action space, $\mathcal{P} : \mathcal{X} \times \mathcal{A} \to \bigwedge(\mathcal{X})$ is the transition function, $\bigwedge(\mathcal{X})$ indicates the next state, $\mathcal{R} : \mathcal{X} \times \mathcal{A} \to [0, 1]$ is the reward function, $\gamma \in [0, 1)$ is a discount factor, and $d_0 \in \bigwedge(\mathcal{X})$ is the distribution of the initial state $x_0$. The goal is to find a policy $\pi : \mathcal{X} \to \bigwedge(\mathcal{A})$ that maximizes the expected cumulative discount bonus $\mathbb{E}_\pi[\sum_{t=0}^{\infty} \gamma^t r_t]$, where $x_0 \sim d_0$, and $\forall t$ we have $a_t \sim \pi(\cdot|x_t)$, $x_{t+1} \sim \mathcal{P}(\cdot|x_t, a_t)$, and $r_t = R(x_t, a_t)$. In pixel observation environments, a per common practice Hosu & Rebedea (2016), called frame-stack, is generally applied.

### 3.2 POSITIVE UNLABELED POLICY-GUIDED CONTRASTIVE REINFORCEMENT LEARNING

PUPG can be considered as an auxiliary task with any RL algorithms for training pixel-based RL. This section discusses three aspects: data augmentation, policy-guided representation, and positive unlabeled contrastive loss. The whole framework of PUPG is shown in Figure 2.

**Data Augmentation**. On the one hand, the semantic information distribution between CV data sets and RL environments is different, where it is more concentrated in CV datasets than in many RL environments. On the other hand, the images augmented by a strong random crop still have semantic information for learning representations. Still, the RL agent can not learn the optimal policy once the semantic information is cropped. See Appendix E of how strong random resized crop disrupts policy training. In our implementation, we apply a weak random resized crop and random plasma shadow to supplement the strength of data enhancement with as little decrease of semantic information as possible.

**Policy-Guided Representation**. Representation learned in pixel-based RL should help to find a better policy instead of only the observation representation. Only observation representation learning leads the policy to a local optimum, even though it may help the policy training in the early stage. We analyze this theoretically in Section 4. To this end, we introduce our policy-guided representation, which can help pixel-based RL to catch more task-related semantic information to the optimal policy. Similar to Haarnoja et al. (2018); Mazoure et al. (2020); Huang et al. (2023), our policy-guided representation is formulated as:

$$Z = (f(s), \pi(s)) = (f(s; \theta), h(f(s; \theta); \phi)) = y \oplus \pi \tag{1}$$

where $f$ is the encoder with its parameters $\theta$, $h$ is the policy network with its parameters $\phi$ and, $\oplus$ stands for concatenation.

**Positive Unlabled Contrastive Learning**. After the policy-guided representation is designed, we pursue an explicit optimization with a criterion that directly measures the structure of representations to minimize the biases brought by contrastive learning. Unlike the samples in CV data sets that are clear of different classes, like a plane to a cat, it is hard to define the negative view to an

anchor view in pixel observation environments. So contrastive learning without negative views is more simple and effective. To this end, we follow the maximum entropy principle. Entropy is defined initially on probability distributions Shannon (1948), i.e., $H(z) \triangleq - \int p(z) \log p(z) dz$, for continuous random variables. However, it is challenging to estimate the true distributions $p(x)$ of a representation Beirlant et al. (1997); Paninski (2003), from a finite set of high dimensional vectors $Z = [z^1, z^2, ..., z^m] \in \mathbb{R}^{d \times m}$ and encoding the data lossless in infeasible in this case Liu et al. (2022). Instead, we exploit the coding length in lossy data coding Cover & Thomas (2006) as a computationally tractable surrogate for the entropy of continuous random variables.Given a set of samples $Z$, the minimal number of bits needed to encode $Z$ subject to a distortion $\epsilon$ is given by the following coding length function Ma et al. (2007); Vidal et al. (2005):

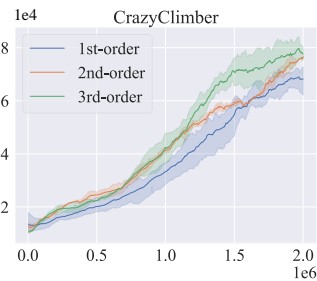

Figure 3: Performance under different expansion orders.

$$L \triangleq (\frac{m+d}{2}) \log \det(I_m + \frac{d}{m\epsilon^2} Z^T Z) \qquad (2)$$

where $I_m$ denotes the identity matrix with dimension $m$, and $\epsilon$ is the upper bound of the expected decoding error between $z \in Z$ and the decoded $\hat{z}$, i.e., $\mathbb{E}||z - \hat{z}||_2 \leq \epsilon$.

The computation of the log-determinant of a high dimensional matrix in Equation 2 suffers from high expense and may cause numerically unstable results for an ill-conditioned matrix. Inspired by Liu et al. (2022), we rewrite Equation 2 as $L = \mu \log \det(I_m + \lambda Z^T Z')$, where $\mu = \frac{m+d}{2}$ and $\lambda = \frac{d}{m\epsilon^2}$, $Z'$ is another view of the sample $Z$. Utilizing the identical equation Horn & Johnson (2012), then it becomes $L = \text{Tr}(\mu \log(I_m + \lambda Z^T Z'))$, where $Tr$ stands for the trace of the matrix. With Taylor series expansion applied to it, we obtain the $1^{st}$-order, $2^{nd}$-order, and $e^{rd}$-order expansion:

$$L = \begin{cases} Tr(\mu(\lambda Z^T Z')), & 1^{\text{st}}\text{-order} \\ Tr(\mu(\lambda Z^T Z') - \frac{\mu}{2}(\lambda Z^T Z')^2), & 2^{\text{nd}}\text{-order} \\ Tr(\mu(\lambda Z^T Z') - \frac{\mu}{2}(\lambda Z^T Z')^2 + \frac{\mu}{3}(\lambda Z^T Z')^3, & 3^{\text{rd}}\text{-order} \end{cases} \qquad (3)$$

Though the results of a toy experiment in Figure 3, the $1^{st}$-order expansion shows the worst outcomes, the $3^{rd}$-order expansion enjoys the best performance while suffering from a costly computation. Although the $2^{nd}$-order expansion does not achieve the best performance, there is only a 2% performance gap to the $3^{rd}$-order expansion. Therefore, we further implement our policy-guided representation in the $2^{nd}$-order expansion in Equation 3, and our positive unlabeled policy-guided contrastive loss can be formulated as:

$$\mathcal{J}_{PUPG} = \mu \underbrace{\sum_{i=1}^{d}(-C_{ii} + \frac{1}{2}C_{ii}^2)}_{\text{Policy-Guided Consistency}} + \frac{\mu}{2} \underbrace{\sum_{i=1}^{d}\sum_{j \neq i}^{d} C_{ij}^2}_{\text{Policy-Guided Informativeness}} \qquad (4)$$

where $C \triangleq \lambda Z' Z^T$. Notably, our policy-guided contrastive loss can be jointly trained with any other pixel-based RL algorithm.

## 4 THEORETICAL INSIGHTS WITH CONNECTION TO INFORMATION THEORY

In this section, we revisit pixel-based reinforcement learning from the information theory perspective and provide analyses of the intrinsic reason behind the performance superiority of the proposed PUPG, which is organized as follows: 1) the connection between the pixel-based RL and the information theory; 2) the interpretation of the performance superiority of proposed PUPG with respect to the information entropy and mutual information. For details, refer to **Appendix** A.

**Notations**. Denote the random variable of input data as $X$, the representation learned from $X$ as $R$, i.e., $R = f(X)$, $R^\star$ as the representation learned by following the paradigm of PUPG, the augmented views of $X$ as $S$ and $S'$, and the downstream policy-optimization task-relevant information as $T$. Note that $T$ presents the information related to the *rewards* of RL, which is orthogonal to the discriminative *visual semantic* information, denoted as $V$, derived by the self-supervised contrastive

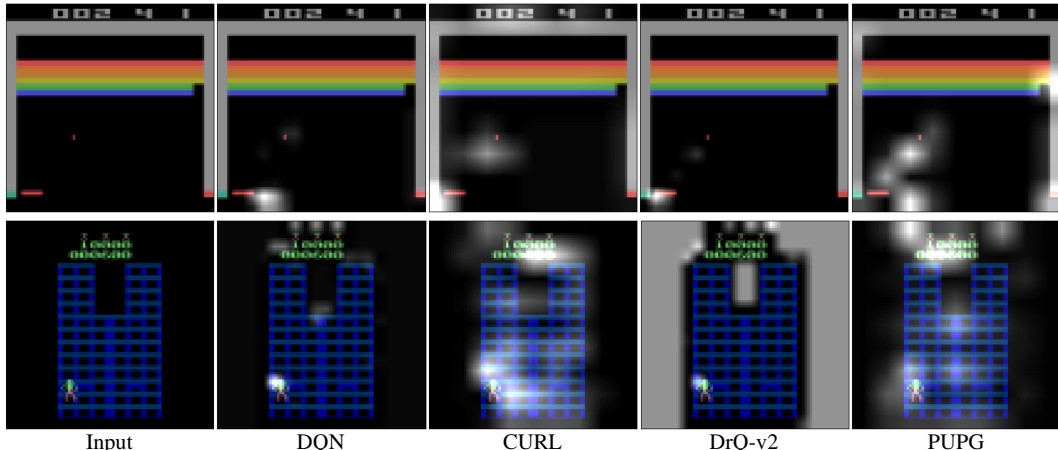

Figure 4: Grad-CAM Selvaraju et al. (2020) visualizations of compared pixel-based RL methods on Breakout (top row) and CrazyClimber (bottom row).

learning, i.e., $T$ and $V$ share an intersection and the corresponding unique complements. For random variables $A$, $B$, and $C$, $I(A, B)$ denotes the mutual information between $A$ and $B$, $I(A, B|C)$ denotes the conditional mutual information of $A$ and $B$ on a certain condition of $C$, $H(A)$ presents the information entropy of $A$, and $H(A|B)$ presents the conditional entropy of $A$ given $B$.

**Information Theoretical Framework**. For building the information theoretical framework of pixel-based RL, we introduce assumptions with sufficient empirically analytical evidence as follows:

**Assumption 1.** *(Task-relevant information variance). Pixel-based RL task-relevant information $T$ is majorly contained by the original pixel observation $X$, and the mechanism behind data augmentation is cropping out specific information and adding noise into the pixel observation, thereby deriving an augmented view of $X$, i.e., $S$. The variance of $T$ between $X$ and $S$ exists. In this regard, the identical equation holds: $I(X;T) - I(S;T) = I(X;T|S)$, where $I(X;T|S)$ is not trivial.*

**Assumption 2.** *(Sparsity of task-relevant information in pixel observation). The pixel observation $X$ contains a wealth of noise, denoted as $\varepsilon^{noise}$, and limited task-relevant information $T$. Formally, $H(X|\varepsilon^{noise}) = I(X;T)$, where $H(\varepsilon^{noise}) \gg I(X;T)$.*

**Assumption 3.** *(Connection between visual semantic information and task-relevant information). The visual semantic information $V$, derived from the pixel observation $X$, is supportive to the objective of RL, i.e., $V$, and the task-relevant information $T$ contains intersections. Formally, $I(T;V) \neq 0$.*

**Assumption 4.** *(Redundancy of pixel-based RL). The existence of redundancy of the representation $R$ learned by conventional pixel-based RL methods is generally determined. Formally, suppose $\lfloor \cdot \rceil^k$ present a function acquiring $k$-th dimension feature vector from a representation, $N^D$ denote the dimensionality of $R$, and there exists $I(\lfloor R \rceil^k; \lfloor R \rceil^m) \neq 0$, where $k, m \in [\![1, N^D]\!]$.*

Assumptions can be sufficiently proved by the empirical evidences. The fact that data augmentation imposes cropping and Gaussian noise on the pixel observation and the empirical proof in Figure 4 that DQN, based on the original pixel observation, acquires representations with more task-relevant information than DrQ-v2, based on the augmented view, are supportive to the validation of Assumption 1. Figure 4 demonstrates that in RL environments, the task-relevant information is only contained by a limited part of the pixel observation, which proves Assumption 2. For Assumption 3, CURL and PUPG exploring visual semantic information learn representations with more task-relevant information than DQN, e.g., the brick status in Breakout and the scores and HP in Crazy-Climber in Figure 4. PUPG, leveraging the informativeness term, can indeed learn more informative representation than benchmark methods, which supports Assumption 4. Refer to **Appendix** A.2 for further empirical evidences and discussions. Based on the solid assumptions, we can build the information theoretical framework for pixel-based RL, illustrated in Figure 5.

**Insights**. As shown in Figure 5, DQN solely trains the encoder by back-propagating the RL loss, resulting in the insufficiency of exploring task-relevant information and the acquirement of exces-

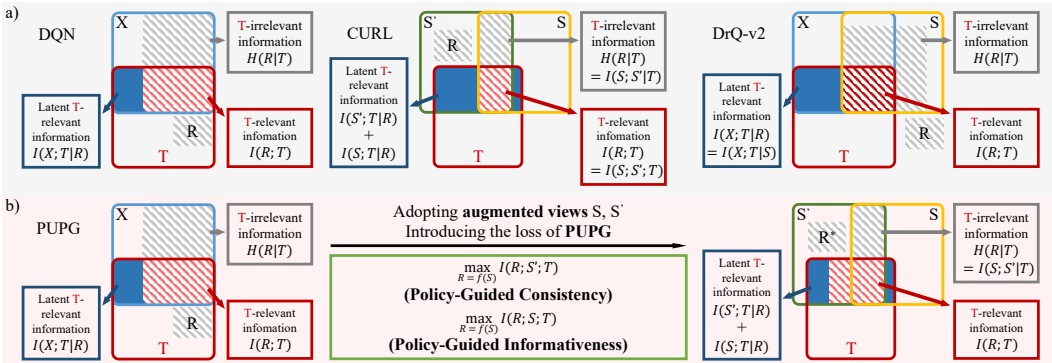

Figure 5: Theoretical illustration towards pixel-based reinforcement learning benchmarks and the proposed PUPG from the information theory perspective, where the sub-figure a) demonstrates the benchmark methods and the sub-figure b) presents PUPG. Note that the shaded parts (red and grey) of the figure depict $R$ and $R^\star$, and $R^\star$ is optimized from $R$ via introducing PUPG.

sive noise due to Assumption 2 and Assumption 4. CURL explores visual semantic information without the guidance of task-relevant information of RL, and DrQ-v2 solely learn representations from the augmented views, incurring certain limitation of the task-relevant information learned by such methods due to Assumption 1. According to Assumption 3, CURL derives less task-irrelevant information than DQN on most benchmarks. The analyses on benchmark are validated by Figure 4. To address the existing issues of benchmark methods, we intuitively propose to learn a policy-aware discriminative representation from the pixel observation as follows:

**Definition 1.** *(Informative pixel-based representation with sufficient policy-optimization-relevant visual semantic information). Let $R$ denote the initial representation learned from the augmented pixel observation $S$ by conventional pixel-based RL method, $S'$ denote another augmented pixel observation adhering the consistent augmented distribution with $S$, and $R^\star$ denote the informative pixel-based representation with sufficient policy-optimization-relevant visual semantic information. Formally, $Y^* = \underset{Y}{argmax}\, I(R; S'; T) + I(R; S; T)$ s.t. $R = f(S)$ where $f(\cdot)$ is the encoder.*

As shown in Figure 5, the objective in Definition 1 can be achieved by the proposed policy-guided consistency and policy-guided informativeness terms. Holding this insight, we theoretically derive

**Theorem 1.** *(Tighter Lower Bound of Mutual Information Guarantee for PUPG). Suppose $R$ denotes the initial representation without optimizing PUPG, and $R^\star$ denotes the representation learned by PUPG. Formally, considering $X$, $S$ and $S'$, $I(X; R; T) - I(X; R|T) \leq I(S; R^\star; T) - I(S; S'; R^\star|T)$, where $R = f(X)$ and $R^\star = f(S)$.*

According to Theorem 1, we conclusively state that without loss of generality, PUPG acquires pertinent representations having tighter lower bound of the mutual information with the policy optimization than benchmark methods. **Appendix** A.1 elaborates the proof of theorem.

## 5 EXPERIMENTS

This section empirically evaluates PUPG on an extensive set of pixel observation environments.

### 5.1 MAIN EXPERIMENTS

We benchmark the performance of PUPG for both discrete and continuous control environments. Specifically, we focus on the Atari benchmark for discrete control tasks and DMControl for continuous control tasks. We compare with several baselines including DQN Hosu & Rebedea (2016), Haarnoja et al. (2018), CURL Laskin et al. (2020b), and DrQ-v2 Yarats et al. (2022). These baselines are competitive methods for benchmarking control from pixel observations. More experiment details are reported in Appendix B and Appendix C.

Table 1: Atari average score after 2M steps training.

| Game | Human | Random | DQN | CURL | DrQ-v2 | PUPG |
|---|---|---|---|---|---|---|
| Alien | 7127.7 | 345.6 | 1159.5 | 811.0 | 1027.5 | **1403.5** |
| Amidar | 1719.5 | 20.5 | 574.0 | 137.8 | 574.8 | **642.0** |
| Assault | 742.0 | 206.2 | 3007.3 | 741.1 | 3138.6 | **3809.2** |
| Asterix | 8503.3 | 257.0 | 2162.5 | 1490.0 | 2567.5 | **2680.0** |
| BankHeist | 47388.7 | 13.5 | **160.0** | 12.0 | 30.5 | 44.0 |
| BattleZone | 37187.5 | 2730.0 | 28850.0 | 13250.0 | 4850.0 | **31050.0** |
| Boxing | 12.1 | -8.8 | 34.8 | **58.6** | 27.7 | 56.6 |
| Breakout | 30.5 | 1.9 | 25.6 | 13.0 | 31.7 | **35.2** |
| ChopperCommand | 7387.8 | 733.0 | 4680.0 | 1980.0 | 6175.0 | **6305.0** |
| CrazyClimber | 35829.4 | 10110.0 | 63155.0 | 51190.0 | 24405.0 | **94450.0** |
| DemonAttack | 1971.0 | 126.2 | 8256.2 | 2558.0 | 9488.8 | **11382.0** |
| Freeway | 29.6 | 0.3 | 31.1 | 24.8 | 27.4 | **31.4** |
| Frostbite | 4334.7 | 65.5 | 2230.5 | 2445.0 | 1846.0 | **4131.5** |
| Gopher | 2412.5 | 180.8 | 1520.0 | 794.0 | 816.0 | **3763.0** |
| Hero | 30826.4 | 1654.0 | 11649.8 | 10823.0 | 7309.8 | **12130.2** |
| IceHockey | 0.9 | -13.2 | -7.0 | -5.8 | -10.4 | **-5.4** |
| Jamesbond | 302.8 | 39.0 | 505.0 | 205.0 | 550.0 | **690.0** |
| Kangaroo | 3035.0 | 22.0 | 9260.0 | 370.0 | 6820.0 | **9550.0** |
| Krull | 2665.5 | 1534.4 | 35792.5 | 4211.0 | 15264.5 | **46669.0** |
| KungFuMaster | 22736.3 | 59.0 | 14430.0 | 1185.0 | 8890.0 | **14590.0** |
| MsPacman | 6951.6 | 475.8 | 2138.5 | 1370.0 | 2273.5 | **2426.0** |
| Pong | 14.6 | -20.7 | 2.9 | -17.5 | -10.0 | **4.5** |
| PrivateEye | 69571.3 | -379.2 | 37.8 | 68.0 | 70.0 | **178.0** |
| Seaquest | 42054.7 | 60.4 | 1717.0 | 1529.0 | 1766.0 | **2313.0** |
| UpNDown | 11693.2 | 1007.0 | **9321.5** | 3623.0 | 5770.0 | 8987.5 |

Generally, our method PUPG outperforms the baselines in 22 of all 25 Games after training 2M steps. We also report the evaluated results on Atari after 500K interaction steps in Appendix D. Comparing CURL and DrQ-v2 perform better than DQN in 6 and 12 games, respectively, they only perform better than DQN in 4 and 10 games. This phenomenon, which is the observation representation learning, is helpful to sample-efficient RL in the early training stage while dragging the training in the later stage, which coincides with our previous analysis. It is more evident in the training curves. As shown in Figure 1 and Appendix F, the CURL and DrQ-v2's curves rise much faster than DQN's in the early stages of training, but there is a high probability that DQN will catch up later. However, PUPG does not have this phenomenon and demonstrates better training results. We also evaluated our method in continuous control environments. The local optimum phenomenon is not apparent since DMControl tasks are more challenging than Atari. Still, our method outperforms the baselines in 5 of 8 tasks.

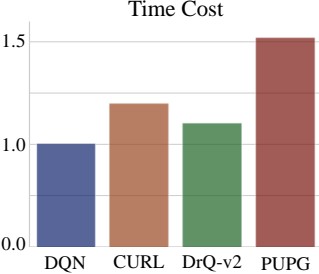

Figure 6: Time cost of each method. DQN as unit.

## 5.2 TIME COMPLEXITY AND SAMPLE-EFFICIENCY EXPERIMENTS

We further analyze the computational complexity of PUPG, comparing the baselines. PUPG has a $O(T \times M)$, which is the same as baselines, where $T$ is the length of an episode and $M$ is the number of episodes. The calculation process is shown in Appendix C.2. In practice, the time cost of PUPG, shown in Figure 6, is about 1.5 times that of DQN.

## 5.3 ABLATION: INFLUENCE OF EACH MODULE

To investigate the properties of PUPG, we set up a series of ablation studies, where we evaluate the PUPG with observation representation (PUPG-OC) and PUPG without random plasma shadow (PUPG-PC). The evaluating results are reported in Figure 8. First at all, The positive unlabeled contrastive loss is not worse than that in CURL. Moreover, policy-guided representation is one of

Table 2: DMControl Average Score after 1M Step

| Task | Acrobot, Swingup | Cartpole, Swingup | Cartpole, Swingup Sparse | Cup, Catch |
|------|------------------|-------------------|--------------------------|------------|
| SAC | 138.36±63.34 | 71.78±35.8 | 164.12±88.68 | 166.13±48.11 |
| CURL | 669.79±114.38 | 756.59±62.89 | 750.58±34.52 | 754.47±26.01 |
| DrQ-v2 | 154.12±38.88 | **851.62±6.68** | 764.02±50.88 | 765.48±38.95 |
| PUPG | **773.78±85.59** | 773.16±46.85 | **850.83±16.61** | **794.63±31.68** |

| Task | Finger, Spin | Finger, Turn Easy | Hopper, Stand | Reacher, Easy |
|------|--------------|-------------------|---------------|---------------|
| SAC | 203.79±62.07 | 92.3±39.12 | 118.55±37.51 | 196.08±67.07 |
| CURL | **950.6±16.72** | 562.12±114.32 | 694.22±62.75 | 554.88±63.29 |
| DrQ-v2 | 734.64±93.71 | 644.44±160.0 | **883.76±42.79** | 955.8±34.46 |
| PUPG | 779.74±57.59 | **720.82±58.31** | 671.15±85.47 | **959.9±33.8** |

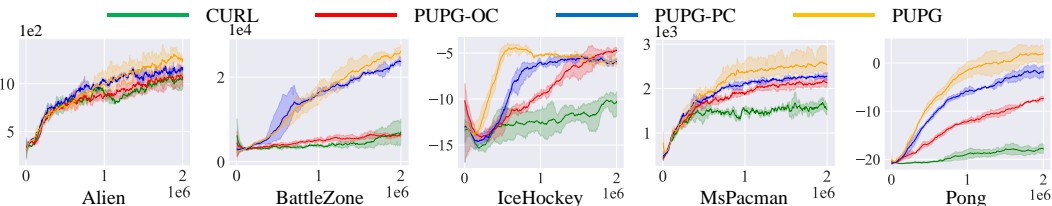

Figure 8: Performance on 5 Atari games

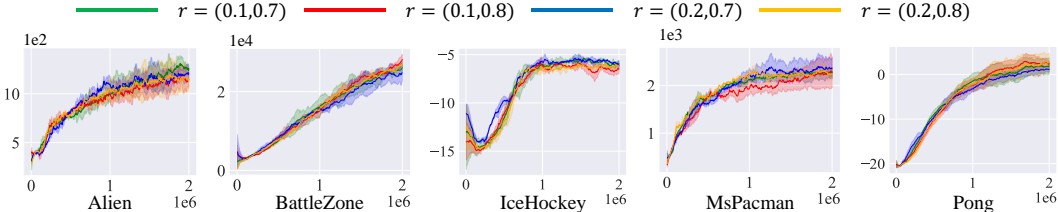

Figure 9: Performance on 5 Atari games

the indispensable modules of PUPG meeting the conclusion discussed in Section 4. In addition, with random plasma shadow applied, strengthening the strength of data augmentation contributes to semantic information learning.

## 5.4 ROBUSTNESS TO THE PARAMETERS RANDOM PLASMA SHADOW

The core parameter of random plasma shadow is roughness, denoted as $r$. We evaluate our method under four sets of parameters. As reported in Figure 9, our method achieves similar results under four sets of parameters, illustrating that our method is robust for the parameters of random plasma shadow, and no additional parameter tuning is needed.

## 6 CONCLUSIONS

In this work, we have proposed PUPG, a positive unlabeled policy-guided contrastive reinforcement learning for pixel-based RL that achieves state-of-the-art sample efficiency on pixel-based RL evaluated on various performance experiments and ablation studies across a diverse set of benchmark environments. We are the first to analyze the representation of pixel-based RL from an information theory perspective. However, our method still suffers from the need for careful tuning on the parameters of visual data augmentation, predominantly random resized crop. Otherwise, it will lead training to local optimum or even ineffective training. All in all, we hope our algorithm will help inspire and democratize further research in pixel-based RL.

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

# A  THEORETICAL PROOFS AND DISCUSSIONS

We detail the proof to support the correctness and integrity of the proposed information theoretical analyses as follows:

## A.1  PROOFS FOR THEOREM

Here, we provide the proof for Theorem 1.

Theorem 1 states that suppose $R$ denotes the initial representation without the optimization of PUPG, and $R^\star$ denotes the representation learned by PUPG. Formally, considering $X$, $S$ and $S'$, $I(X;R;T) - I(X;R|T) \leq I(S;R^\star;T) - I(S;S';R^\star|T)$, where $R = f(X)$ and $R^\star = f(S)$. We validate that $I(X;R;T) - I(X;R|T) \leq I(S;R^\star;T) - I(S;S';R^\star|T)$ by introducing the estimation of lower bound of mutual information via adopting the KL-divergence Leibler (1951) measurement as follows:

*Proof.* To proof $I(X;R;T) - I(X;R|T) \leq I(S;R^\star;T) - I(S;S';R^\star|T)$

Transpose the mentioned equation into:

$$I(X;R;T) \leq I(S;R^\star;T), \tag{5}$$

and

$$I(X;R|T) \geq I(S;S';R^\star|T). \tag{6}$$

To demonstrate the target inequalities, we introduce the estimation of mutual information as follows:

Suppose $I(A;B)$ denote the mutual information of $A$ and $B$, and we have

$$
\begin{aligned}
I(A;B) &= \sum_{a \in A} \sum_{b \in B} \mathcal{P}(a,b) \log \frac{\mathcal{P}(a,b)}{\mathcal{P}(a) \cdot \mathcal{P}(b)} \\
&= \int \int \mathcal{P}(a,b) \log \frac{\mathcal{P}(a,b)}{\mathcal{P}(a) \cdot \mathcal{P}(b)} da\, db \\
&= D_{KL}(\mathcal{P}_{AB} || \mathcal{P}_A \mathcal{P}_B),
\end{aligned}
\tag{7}
$$

which is a particular form of F-divergence Sason & Verdú (2016), and F-divergence can be formalized by

$$D_F(\mathcal{P}_A || \mathcal{P}_B) = \int \int \mathcal{P}(b) g\left(\frac{\mathcal{P}(a)}{\mathcal{P}(b)}\right) da\, db, \tag{8}$$

where $g(\cdot)$ is a projection function, s.t., $g(\cdot)$ adheres the closed convex property and $g(1) = 0$.

Then, we have

$$D_{KL}(\mathcal{P}_{AB} || \mathcal{P}_A \mathcal{P}_B) = D_F(\mathcal{P}_A || \mathcal{P}_B), \tag{9}$$

when $g(x) = x \log x$.

For In-equation 5, we notice that $R = f(X)$ and $R^\star = f(S)$, and thus

$$I(X;R;T) = I(R;T), \tag{10}$$

and

$$I(S;R^\star;T) = I(R^\star;T), \tag{11}$$

which can be derived by following the Data Processing Inequality Thomas et al. (1991).

Thus, we can prove In-equation 5 by introducing Equation 8 as follows:

$$D_F(\mathcal{P}_R || \mathcal{P}_T) = \int \int \mathcal{P}(t) g\left(\frac{\mathcal{P}(r)}{\mathcal{P}(t)}\right) dr\, dt. \tag{12}$$

Considering the desired property of $g(\cdot)$, i.e., $g(\cdot)$ is a closed convex function, we transform Equation 12 into

$$
\begin{aligned}
D_F(\mathcal{P}_R||\mathcal{P}_T) &= \int \int \mathcal{P}(t) g\left(\frac{\mathcal{P}(r)}{\mathcal{P}(t)}\right) dr dt \\
&= \int \int \mathcal{P}(t) \left(\max_{q \in dom(g^\star)} \left\{\frac{\mathcal{P}(r)}{\mathcal{P}(t)} \cdot q - g^\star(q)\right\}\right) dr dt,
\end{aligned}
\tag{13}
$$

which is achieved by replace $g(\cdot)$ with the corresponding conjugate bifunction $g^{\star\star}(\cdot)$. In the aforementioned equation, $g^\star(\cdot)$ denotes the conjugate function of $g(\cdot)$, and $dom(\cdot)$ is a function deriving the values in the defined area of the domain.

Then, suppose $Q^R(\cdot)$ and $Q^T(\cdot)$ are arbitrary projection functions with the input of $r$ or $t$ and the output of $q$, respectively, and we can derive

$$
\begin{aligned}
D_F(\mathcal{P}_R||\mathcal{P}_T) &\geq \int \int \mathcal{P}(t) \left(\frac{\mathcal{P}(r)}{\mathcal{P}(t)} Q^R(r) Q^T(t) - g^\star\left(Q^R(r) Q^T(t)\right)\right) dr dt \\
&= \int \mathcal{P}(r) \cdot Q^R(r) dr - \int \mathcal{P}(t) \cdot g^\star\left(Q^T(t)\right) dt \\
&= \max_{Q^R, Q^T} \left\{\int \mathcal{P}(r) \cdot Q^R(r) dr - \int \mathcal{P}(t) \cdot g^\star\left(Q^T(t)\right) dt\right\} \\
&= \max_{Q^R, Q^T} \left\{\mathbb{E}_{r \sim \mathcal{P}_R}\left[\mathcal{P}(r) \cdot Q^R(r)\right] - \mathbb{E}_{t \sim \mathcal{P}_T}\left[\mathcal{P}(t) \cdot g^\star\left(Q^T(t)\right)\right]\right\},
\end{aligned}
\tag{14}
$$

which represents the optimal estimated lower bound of $D_F(\mathcal{P}_R||\mathcal{P}_T)$, i.e., the optimal estimation of the lower bound of the mutual information $I(R;T)$. Such a deduction is achieved by performing the optimization and thus deriving the optimal $Q^R(\cdot)$ and $Q^T(\cdot)$ towards the maximization of the estimated lower bound of $D_F(\mathcal{P}_R||\mathcal{P}_T)$.

Deductively evidenced by the same token from Equation 7 to Equation 15, we derive the estimation of the lower bound of the mutual information $I(R^\star;T)$ as follows:

$$
\begin{aligned}
I(S; R^\star; T) &= I(R^\star; T) \\
&= D_F(\mathcal{P}_{R^\star}||\mathcal{P}_T) \\
&= \max_{Q^{R^\star}, Q^T} \left\{\mathbb{E}_{r \sim \mathcal{P}_{R^\star}}\left[\mathcal{P}(r) \cdot Q^{R^\star}(r)\right] - \mathbb{E}_{t \sim \mathcal{P}_T}\left[\mathcal{P}(t) \cdot g^\star\left(Q^T(t)\right)\right]\right\}.
\end{aligned}
\tag{15}
$$

Thus, the proof of In-equation 5 is transformed into

$$
\mathbb{E}_{r \sim \mathcal{P}_{R^\star}}\left[\mathcal{P}(r) \cdot Q^{R^\star}_{max}(r)\right] \geq \mathbb{E}_{r \sim \mathcal{P}_R}\left[\mathcal{P}(r) \cdot Q^R_{max}(r)\right],
\tag{16}
$$

and

$$
\mathbb{E}_{t \sim \mathcal{P}_T}\left[\mathcal{P}(t) \cdot g^\star\left(Q^T_{max}(t)\right)\right] \leq \mathbb{E}_{t \sim \mathcal{P}_T}\left[\mathcal{P}(t) \cdot g^\star\left(Q^T_{max}(t)\right)\right].
\tag{17}
$$

Due to the inherent property of $Q^R(\cdot)$ and $Q^T(\cdot)$, the expectations of In-equation 16 and In-equation 17 can be approximated by the amount of the discrete task-relevant feature set, derived by the learned representation. Thus, according to Assumption 2, Assumption 3, Assumption 4 and the empirical results, we confidently state that the amount of the discrete task-relevant feature set derived by $R^\star$ is superior to $R$, such that the In-equation 5 holds.

For In-equation 6, we firstly hold that suppose $I(A; B|C)$ denote the conditional mutual information of $A$ and $B$ given $C$, and we have

$$
I(A; B|C) = \sum_{a \in A} \sum_{b \in \{B - C\}} \mathcal{P}(a, b) \log \frac{\mathcal{P}(a, b)}{\mathcal{P}(a) \cdot \mathcal{P}(b)}
\tag{18}
$$

From the deduction from Equation 7 to In-equation 16 and In-equation 17, we derive

$$
\mathbb{E}_{r \sim \mathcal{P}_{R^\star - T}}\left[\mathcal{P}(r) \cdot Q^{R^\star - T}_{max}(r)\right] \geq \mathbb{E}_{r \sim \mathcal{P}_{R - T}}\left[\mathcal{P}(r) \cdot Q^{R - T}_{max}(r)\right],
\tag{19}
$$

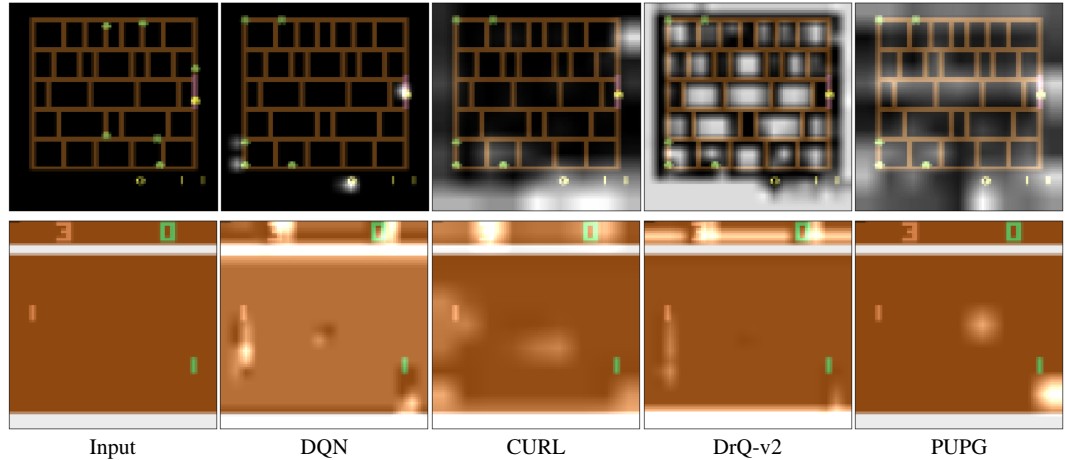

| Input | DQN | CURL | DrQ-v2 | PUPG |

Figure 9: Grad-CAM visualizations of compared pixel-based RL methods on Amidar (top row) and Pong (bottom row).

and

$$\underset{t \sim \mathcal{P}_{R^\star - T}}{E} \left[ \mathcal{P}(t) \cdot g^\star \left( Q_{max}^{R^\star - T}(t) \right) \right] \leq \underset{t \sim \mathcal{P}_{R-T}}{E} \left[ \mathcal{P}(t) \cdot g^\star \left( Q_{max}^{R-T}(t) \right) \right]. \tag{20}$$

Thus, according to the above analyses, we can derive that the amount of the discrete task-relevant feature set derived by $R-T$ is superior to $R^\star - T$ due to Assumption 2 and the empirical results, such that the In-equation 6 holds. In summary, the Theorem 1 is theoretically and empirically validated, i.e.,

$$I(X; R; T) - I(X; R|T) \leq I(S; R^\star; T) - I(S; S'; R^\star|T). \tag{21}$$

$\square$

### A.2   FURTHER EMPIRICAL EVIDENCES AND DISCUSSIONS

As demonstrated in Figure 9, we provide further empirical evidences on various benchmark environments, which are supportive to the validation of the proposed assumptions in Section 4. Specifically, we find the empirical observation consistent in Figure 4 and Figure 9. Thus, the following statement still holds the fact that data augmentation imposes cropping and Gaussian noise on the pixel observation, and the empirical proof in Figure 9 that DQN, based on the original pixel observation, acquires representations with more task-relevant information than DrQ-v2, based on the augmented view, are supportive to the validation of Assumption 1. Figure 9 demonstrates in RL environments, the task-relevant information is only contained by a limited part of the pixel observation, which proves Assumption 2. For Assumption 3, CURL and PUPG exploring visual semantic information learn representations with more task-relevant information than DQN, e.g., the brick status in Breakout and the scores and HP in CrazyClimber in Figure 9. PUPG, leveraging the informativeness term, can learn more informative representation than benchmark methods, which supports Assumption 4.

## B   NETWORK STRUCTURES

The network structures are shown in Table 3. DQN, CURL, and DrQ-v2 only have the backbone and q networks. CURL contrast features on the outputs of the backbone network. PUPG has all three network parts and contrast features on the projector's output.

Table 3: Network Strucutres for Atari

| **Backbone Network** |
|---|
| Conv2d(4, 32, kernel_size=(8, 8), stride=(4, 4)) |
| ReLU(inplace=True) |
| Conv2d(32, 64, kernel_size=(4, 4), stride=(2, 2)) |
| ReLU(inplace=True) |
| Conv2d(64, 64, kernel_size=(3, 3), stride=(1, 1)) |
| ReLU(inplace=True) |
| Flatten(start_dim=1, end_dim=-1) |
| Linear(in_features=3136, out_features=512, bias=True) |
| ReLU(inplace=True) |

| **Q Network** |
|---|
| Linear(in_features=512, out_features=128, bias=True) |
| ReLU(inplace=True) |
| Linear(in_features=128, out_features=action_shape, bias=True) |

| **Projector** |
|---|
| Linear(in_features=521, out_features=512, bias=False) |
| ReLU(inplace=True) |
| Linear(in_features=512, out_features=512, bias=False) |

### B.1 FOR ATARI

### B.2 FOR DMCONTROL TASKS

The network structures are shown in Table 4. Note that SAC, CURL, and DrQ-v2 only have backbone, actor, and critic networks. CURL contrast features on the outputs of the backbone network. PUPG has all four network parts and contrasts features on the projector's output.

## C HYPER-PARAMETERS

The hyper-parameters for Atari and DMControl are reported in Table 5 and Table 6, respectively.

### C.1 CODE BASED

Our code is based on the open-source repository CleanRL. `https://github.com/vwxyzjn/cleanrl` for Atari, and DrQ-v2 `https://github.com/facebookresearch/drqv2` for DMControl tasks.

### C.2 ALGORITHM AND TIME COST

The computation cost is

$$
\begin{aligned}
T(episode, t) = \ & t_0 + t_1 + \\
& (t_2 + t_{2.1} + (t_{2.2.1} + t_{2.2.2} + t_{2.2.3} + ... + t_{2.2.10}) \times T) \times M \\
= \ & t_{c1} + (t_{c2} + t_{c3} \times T)M \\
= \ & t_{c1} + (t_{c2} \times M + t_{c3} \times T \times M) \\
= \ & t_{c1} + t_{c2} \times M + t_{c3} \times T \times M \\
= \ & t_{c3} \times T \times M \\
= \ & T \times M
\end{aligned}
\tag{22}
$$

### C.3 PUPG PSEUDOCODE (PYTORCH-LIKE)

```
1  # f, g, p: encoder, projector, policy network respectively
2  # obs: pixel observation sampled from memory buffer
```

Table 4: Network Strucutres for DMControl Tasks

| **Backbone Network** |
|:---:|
| Conv2d(9, 32, kernel_size=(3, 3), stride=(2, 2)) |
| ReLU(inplace=True) |
| Conv2d(32, 32, kernel_size=(3, 3), stride=(1, 1)) |
| ReLU(inplace=True) |
| Conv2d(32, 32, kernel_size=(3, 3), stride=(1, 1)) |
| ReLU(inplace=True) |
| Conv2d(32, 32, kernel_size=(3, 3), stride=(1, 1)) |
| ReLU(inplace=True) |
| Flatten(start_dim=1, end_dim=-1) |
| Linear(in_features=3136, out_features=512, bias=True) |
| ReLU(inplace=True) |

| **Actor Network** |
|:---:|
| Linear(in_features=39200, out_features=50, bias=True) |
| ReLU(inplace=True) |
| Linear(in_features=50, out_features=1024, bias=True) |
| ReLU(inplace=True) |
| Linear(in_features=1024, out_features=1024, bias=True) |
| ReLU(inplace=True) |
| Linear(in_features=1024, out_features=action_shape, bias=True) |

| **Critic Network** |
|:---:|
| Linear(in_features=39200, out_features=50, bias=True) |
| ReLU(inplace=True) |
| Linear(in_features=50, out_features=1024, bias=True) |
| ReLU(inplace=True) |
| Linear(in_features=1024, out_features=1024, bias=True) |
| ReLU(inplace=True) |
| Linear(in_features=1024, out_features=action_shape, bias=True) |

| **Projector** |
|:---:|
| Linear(in_features=521, out_features=512, bias=False) |
| ReLU(inplace=True) |
| Linear(in_features=512, out_features=512, bias=False) |

```
3   obs_anc = aug(obs)
4   obs_pos = aug(obs)
5   h_anc = f(obs_anc)
6   h_pos = f(obs_pos)
7   a_anc = p(h_anc)
8   a_pos = p(h_pos)
9   z_anc = g(cat((h_anc, a_anc)))
10  z_pos = g(cat((h_pos, a_pos)))
11  c = z_anc.T @ z_pos
12  on_diag = torch.diagonal(c).add_(-1).pow_(2).sum()
13  off_diag = off_diagonal(c).pow_(2).sum()
14  loss = on_diag + self.args.lambd * off_diag
15  loss.backward()
16  update(f.params, g.params, p.params)
```

## D  PERFORMANCE EVALUATED ON ATARI 500K BENCHMARK

The results is shown in Figure 7.

Table 5: Hyper-Parameters for Atari

| Hyper-Pararmeter | Value |
|---|---|
| frame stack | 4 |
| buffer size | 100000 |
| learning rate | 1e-4 |
| start epsilon | 1 |
| end epsilon | 0.01 |
| exploration fraction | 0.05 |
| learning starts step | 5000 |
| batch size | 32 |
| gamma | 0.99 |
| tau | 1 |
| target network update frequency | 1000 |
| train frequency | 4 |
| lambda | 0.0051 |
| random resized crop scale | 0.8 |
| cl coefficiency | 1e-2 |

Table 6: Hyper-Parameters for DMControl Tasks

| Hyper-Pararmeter | Value |
|---|---|
| frame stack | 3 |
| action repeat | 2 |
| gamma | 0.99 |
| evaluate frequency | 10000 |
| evaluate episodes | 10 |
| buffer size | 100000 |
| learning rate | 1e-4 |
| learning starts step | 5000 |
| batch size | 256 |
| gamma | 0.99 |
| critic target tau | 0.01 |
| target network update frequency | 2 |
| train frequency | 4 |
| random resized crop scale | 0.8 |
| cl coefficiency | 1e-2 |
| stddev clip | 0.3 |

---

**Algorithm 1** PUPG: Positive Unlabeled Policy-Guided Contrastive Reinforcement Learning (DQN version)

---

Initialize memory buffer $\mathcal{D}$ to capacityu $N$ (time cost $t_0$)
Initialize backbone network $\{(\cdot; \theta)$, action-value function $\mathcal{Q}(\cdot; \phi)$, which is instantiation of $h$, and projector $g(\cdot; \varphi)$ with random weights (time cost $t_1$)
**for** $episode = 1, M$ **do**
  Initialise sequence $x_1$ (time cost $t_{2.1}$)
  **for** $t = 1, T$ **do**
    With probability $\epsilon$ select a random action $a_t$ (time cost $t_{2.2.1}$)
    Otherwise embed pixel observation by backbone network $y_t = f(x_t; \theta)$ and select $a_t = \max_a Q^\star(y_t, a; \phi)$ (time cost $t_{2.2.1}$ because it is a otherwise situation)
    Execute action $a_t$ in environment and observe reward $r_t$ and pixel observation $x_{t+1}$ (time cost $t_{2.2.2}$)
    Store transition $(x_t, a_t, r_t, x_{t+1})$ in $\mathcal{D}$ (time cost $t_{2.2.3}$)
    Sample random mini-batch of transitions $(x_t, a_t, r_t, x_{t+1})$ (time cost $t_{2.2.4}$)
    Augment $x_t$ and $x_{t+1}$ to $s_t, s_t', s_t^\star$ (time cost $t_{2.2.5}$)
    Obtain policy $Q_t, Q_t'$ of $s_t, s_t'$ (time cost $t_{2.2.6}$)
    Concatenate $f(s_t; \theta) \oplus Q_t$ and $f(s_t'; \theta) \oplus Q_t'$ to $z_t$ and $z_t'$ (time cost $t_{2.2.7}$)
    Calculate contrastive loss by Equation 4 (time cost $t_{2.2.8}$)
    Calculate dqn loss by Hosu & Rebedea (2016) (time cost $t_{2.2.9}$)
    Backward gradient (time cost $t_{2.2.10}$)
  **end for**
**end for**

---

## E  TOY EXPERIMENTS: HOW DOES RANDOM RESIZED CROP HURT TRAINING

In this section, we aim to address the semantic information disruption brought by random resized crop. Three environments are selected, where the semantic information of Breakout and Pong is distributed at the edge of the pixel observation, and BasicMath is concentrated in the center of the pixel observation. We highlight some of the task-related semantic information areas in red rectangles. Then, we define three task-related semantic information lost situations:

1. Nothing Lost: No red area is cut off, and all the task-related semantic information is preserved.

2. Partially Lost: Some area is cut off, and some task-related semantic information is lost.

Table 7: Average Score Performance on Atari 500K Benchmark

| Game | Human | Random | DQN | CURL | DrQ-v2 | PUPG |
|------|-------|--------|-----|------|--------|------|
| Alien | 7127.7 | 345.6 | 662.5 | 722.0 | 486.5 | **781.5** |
| Amidar | 1719.5 | 20.5 | 68.4 | 101.4 | 242.9 | **351.2** |
| Assault | 742.0 | 206.2 | 1165.6 | 328.61 | **1460.8** | 1413.6 |
| Asterix | 8503.3 | 257.0 | 855.0 | 637.5 | **1092.5** | 877.5 |
| BankHeist | 47388.7 | 13.5 | **31.0** | 9.5 | 10.5 | 30.5 |
| BattleZone | 37187.5 | 2730.0 | 7600.0 | 1550.0 | 2350.0 | **12650.0** |
| Boxing | 12.1 | -8.8 | -2.4 | **27.2** | 1.0 | -1.4 |
| Breakout | 30.5 | 1.9 | 22.8 | 2.2 | 24.4 | **27.2** |
| ChopperCommand | 7387.8 | 733.0 | 1115.0 | **1490.0** | 1350.0 | 1330.0 |
| CrazyClimber | 35829.4 | 10110.0 | 19940.0 | **38965.0** | 3305.0 | 28960.0 |
| DemonAttack | 1971.0 | 126.2 | 4005.8 | 98.2 | **5702.5** | 4591.8 |
| Freeway | 29.6 | 0.3 | 25.2 | 9.2 | 21.5 | **25.9** |
| Frostbite | 4334.7 | 65.5 | 741.0 | 340.5 | 647.5 | 644.5 |
| Gopher | 2412.5 | 180.8 | 433.0 | 408.0 | 280.0 | **705.0** |
| Hero | 30826.4 | 1654.0 | 2120.8 | 4299.0 | 510.0 | **6111.8** |
| IceHockey | 0.9 | -13.2 | -11.2 | -13.8 | **-10.1** | -10.2 |
| Jamesbond | 302.8 | 39.0 | 47.5 | 35.0 | **372.5** | 137.5 |
| Kangaroo | 3035.0 | 22.0 | 6070.0 | 130.0 | 940.0 | **7555.0** |
| Krull | 2665.5 | 1534.4 | 8242.5 | 713.5 | 1419.0 | **19121.0** |
| KungFuMaster | 22736.3 | 59.0 | **5540.0** | 1005.0 | 2745.0 | 5130.0 |
| MsPacman | 6951.6 | 475.8 | **1966.0** | 1660.0 | 1289.5 | 1857.5 |
| Pong | 14.6 | -20.7 | -9.8 | -20.4 | -19.8 | **-6.6** |
| PrivateEye | 69571.3 | -379.2 | -85.0 | -90.8 | **885.0** | 100.8 |
| Seaquest | 42054.7 | 60.4 | 247.0 | 164.0 | 338.0 | **451.0** |
| UpNDown | 11693.2 | 1007.0 | 3695.0 | 774.5 | **5946.0** | 5119.5 |

3. Totally lost: All the red rectangle areas are cut off, and no task-related semantic information is left in the pixel observation. The agent cannot make the right decision by this kind of pixel observation.

In practice, we set three environments' coordinates of the red areas as shown in 8. The original pixel observation has a size of $160 \times 210$. The crop strength set is $s = [0.005, 0.2, 0.4, 0.6, 0.8, 1.0]$, which means the area of the cropped pixel observation is $s_i \times 160 \times 210$. We run 500K times random crop, and the results are reported in Table 9.

| Breakout | Area |
|----------|------|
| bricks | [8, 57], [152, 57], [8, 93], [151, 93] |
| board | [62, 189], [78, 189], [62, 192], [78, 192] |
| ball | [93, 152], [95, 152], [93, 155], [95, 155] |

| Pong | Area |
|------|------|
| left board | [16, 149], [20, 149], [16, 165], [20, 167] |
| right board | [140, 44], [144, 44], [140, 60], [144, 60] |
| ball | [33, 146], [34, 146], [33, 150], [34, 150] |

| BasicMath | Area |
|-----------|------|
| first number | [48, 54], [83, 54], [48, 80], [83, 80] |
| second number | [48, 92], [83, 92], [48, 118], [83, 118] |
| line | [48, 120], [84, 120], [48, 122], [84, 122] |
| last number | [48, 132], [83, 132], [48, 158], [83, 158] |

Table 8: Caption

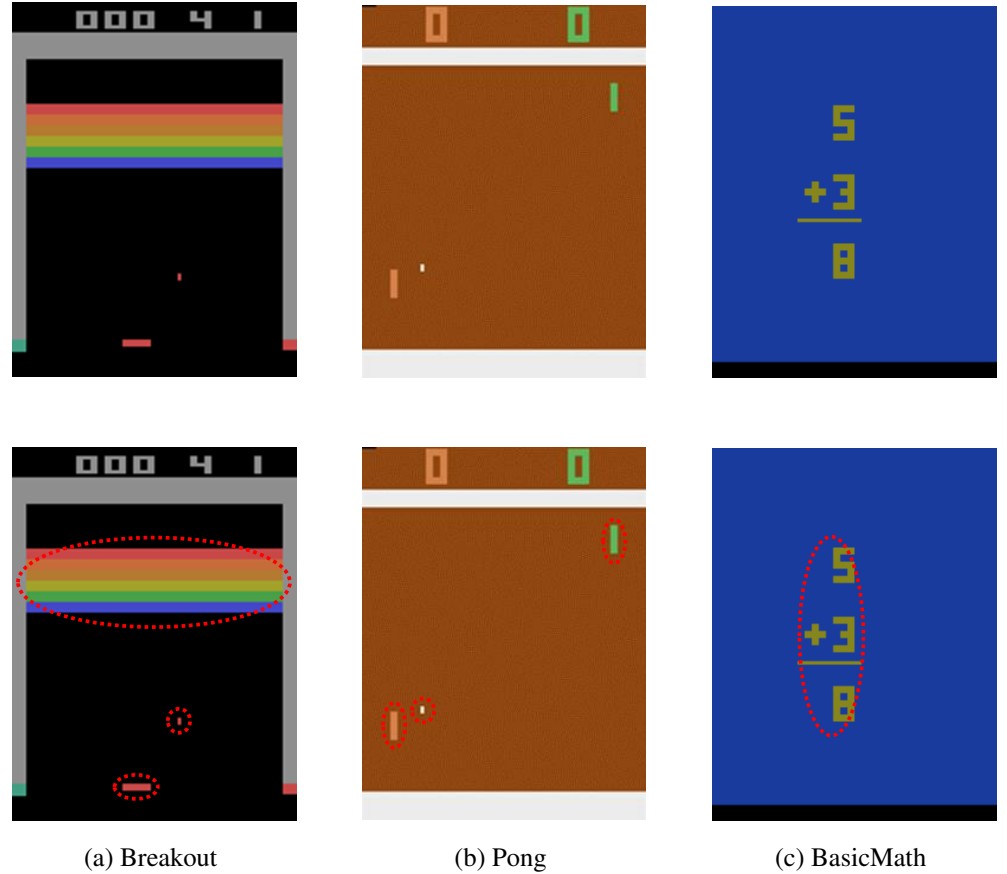

(a) Breakout             (b) Pong             (c) BasicMath

Figure 10: Examples of task-related semantic information areas.

We conduct experiments under different random resized crop strengths at $[0.2, 0.4, 0.6, 0.8, 1.0]$, and the performance decreases as the strength increases. The training curves are reported in Figure 11.

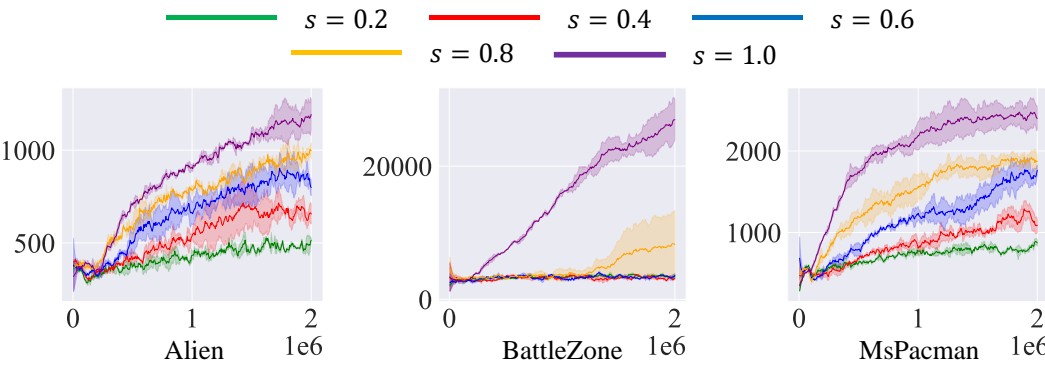

Figure 11: Performance on 5 Atari games.

## F    TRAINING CURVES

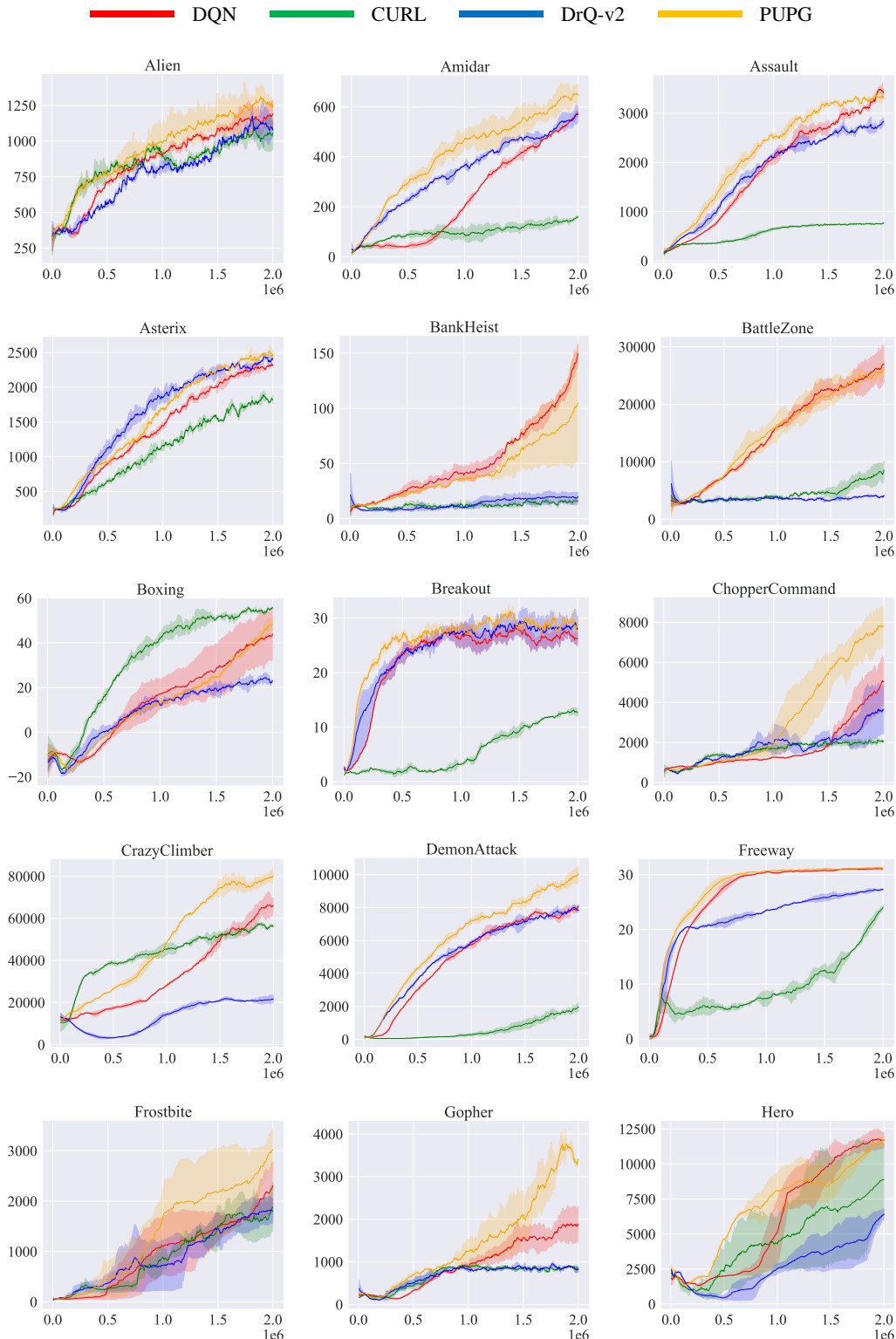

Figure 12: Training curves on 25 Atari games.

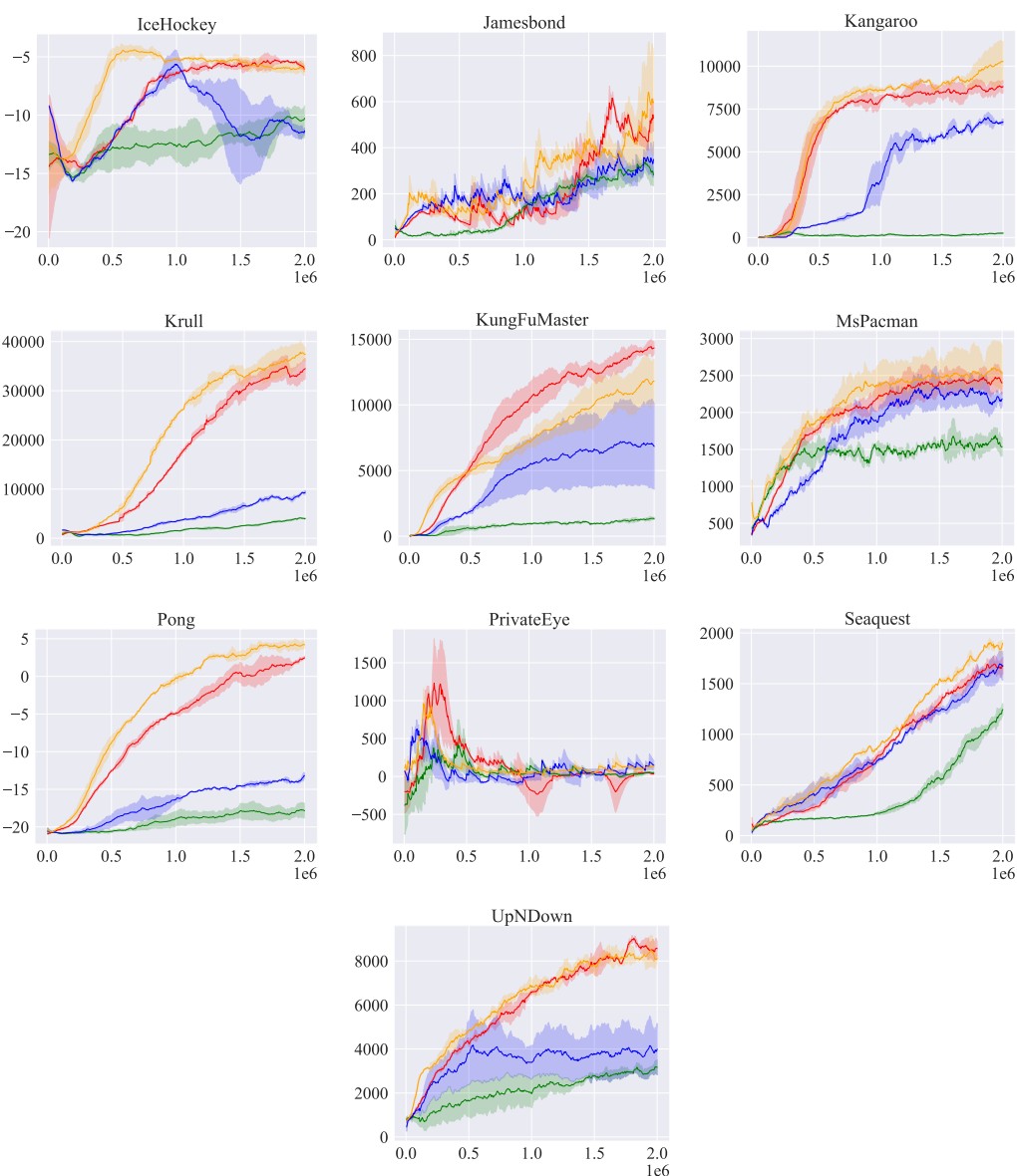

Figure 13: Training curves on 25 Atari. (Continuation of Table 12)

Table 9: Results of different strength of random crop.

| Breakout | Nothing Lost | Partially Lost | Totally Lost |
|---|---|---|---|
| 0.005 | 14.8% | 56.8% | 28.4% |
| 0.2 | 18.5% | 65.9% | 15.6% |
| 0.4 | 24.8% | 70.0% | 5.1% |
| 0.6 | 37.0% | 62.6% | 0.4% |
| 0.8 | 66.6% | 33.4% | 0.0% |
| 1.0 | 100.0% | 0.0% | 0.0% |

| Pong | Nothing Lost | Partially Lost | Totally Lost |
|---|---|---|---|
| 0.005 | 25.9% | 45.2% | 28.9% |
| 0.2 | 32.6% | 52.0% | 15.3% |
| 0.4 | 43.2% | 51.9% | 4.8% |
| 0.6 | 64.3% | 35.5% | 0.2% |
| 0.8 | 93.7% | 6.3% | 0.0% |
| 1.0 | 100.0% | 0.0% | 0.0% |

| BasicMath | Nothing Lost | Partially Lost | Totally Lost |
|---|---|---|---|
| 0.005 | 49.5% | 25.6% | 24.9% |
| 0.2 | 61.8% | 27.7% | 10.5% |
| 0.4 | 79.6% | 17.0% | 3.4% |
| 0.6 | 96.2% | 3.5% | 0.3% |
| 0.8 | 100.0% | 0.0% | 0.0% |
| 1.0 | 100.0% | 0.0% | 0.0% |

