# OpenReview forum: "Mastering Pixel-Based Reinforcement Learning via Positive Unlabeled Policy-Guided Contrast"
_ICLR.cc/2024/Conference — ICLR 2024 Conference Withdrawn Submission_

### Official Review · Reviewer_LKxU · 2023-10-31

**Soundness:** 1 poor
**Presentation:** 1 poor
**Contribution:** 1 poor
**Rating:** 1
**Confidence:** 4

**Summary:**

This paper claims it proposes a new plug-and-play positive unlabeled policy-guided contrast method for pixel-based reinforcement learning. The proposed method learns anti-redundant and policy-optimization-relevant visual semantic information for the training of the RL model. It also contains a theoretical analysis from the information theory perspective, trying to give a closer look at the learning paradigm of pixel-based RL. Experiments are conducted on multiple Atari and DMControl environments.

**Strengths:**

This paper fully complies with the page limit.

**Weaknesses:**

1. The overall presentation of this paper needs to be further improved. For example, though the abstract provides many words, it gives limited information regarding the method or contribution.

2. It will be better to give further elaboration on the method section. For example, in equitation 1, the definitions of y, and \pi are missing, as well as how the concatenation is performed. Meanwhile, I kindly ask for a detailed explanation of the term `policy-guided`.
Similarly, in the paragraph `Positive Unlabled Contrastive Learning.` (there is a typo `Unlabled` in the title as well), a detailed explanation is highly appreciated considering this section should cover the main technical contribution of this paper. There are other terms whose definitions are not clearly presented: `over-redundancy` in the abstract, `local optimum` in Fig.1 and introduction, and so on.

3.  In section 4, the paper says
> Note that T presents the information related to the rewards of RL, which is orthogonal to the discriminative visual semantic information, denoted as V

This is a strong claim that requires firm evidence (which can not be found in the paper). A detailed explanation of the rest of section 4 is highly appreciated.

4. Regarding the experiment settings: for sample-efficient RL, 2M training steps for Atari and 1M for DMControl are quite larger than a typical setting which is 100k. The number of random seeds under the same setting is also missing. The authors are highly encouraged to provide the reward-step curve for DMControl experiments.
The author also mentioned they applied random plasma shadow as data augmentation, Is the augmentation applied to other baselines as well?

5. Jointly optimizing the self-supervised loss with RL loss for pixel-based RL has been extensively explored in [1] and it seems [1] does not give a positive answer even with an evolutionary search on the combination of SSL losses. Does the improvement really come from the new method proposed in this paper, or only from the random plasma shadow?

6. The reward curves in Figure 1 do not contain any axis labels.
7. There are two Figure 9.

### Reference
[1] Li, Xiang, et al. "Does self-supervised learning really improve reinforcement learning from pixels?." Advances in Neural Information Processing Systems 35 (2022): 30865-30881.

**Questions:**

Mentioned in the weakness section.

---

### Official Review · Reviewer_E7VW · 2023-11-01

**Soundness:** 3 good
**Presentation:** 3 good
**Contribution:** 2 fair
**Rating:** 5
**Confidence:** 4

**Summary:**

This paper highlights the paradox of how exploring visual semantic information can simultaneously improve and hinder learned feature representations during different training phases. The study identifies the over-redundancy issue as a limitation in baseline methods, and proposes a novel plug-and-play method to address this deficiency. The method utilizes positive unlabeled policy-guided contrast to learn anti-redundant and policy-optimization-relevant visual semantic information. The effectiveness of the proposed method is supported by theoretical evidence and extensive benchmark experiments, demonstrating superior performance in pixel observation environments compared to existing methods.

**Strengths:**

+ The paper is well written.
+ The structure is coherent and well-arranged, with sentences flowing seamlessly.
+ The technical detail is clear and easy to understand.
+ The theoretical discussion is enriched by intriguing mathematical perspectives。

**Weaknesses:**

1. The author claims that "in different training phases, exploring visual semantic information can both enhance and impede the performance of learned feature representations." I would like to know when this exploration can lead to enhancement, and when it can prevent improvement.

2. The author seems to have different viewpoints compared to SODA. In SODA, the author believes that as more factors of variation are introduced during training, optimization becomes increasingly challenging, which can lead to lower sample efficiency and unstable training. Therefore, they propose decoupling augmentation from policy learning. However, the author of the current study argues that the contrast-based training paradigm is decoupled from the policy network, which interferes with the training of the RL model and ultimately degrades its performance. Regarding the performance of PUPG under strong augmentation methods like random convolution, it is not explicitly mentioned in the provided text.

3. Is there any quantitative evidence to demonstrate that redundant representations indeed lead to a reduction?

4. Why is it said that DQN acquires representations with more task-relevant information than DrQ-v2?

5. Many related methods have not been compared and analyzed. For example, the DrAC approach eliminates redundant representations by constraining the policy to be invariant to the augmented and non-augmented images. How does the author's approach compare to this method in terms of advantages?
[1] Automatic Data Augmentation for Generalization in Reinforcement Learning.
[2] Action-driven contrastive representation for reinforcement learning.
[3] TACO: Temporal Latent Action-Driven Contrastive Loss for Visual Reinforcement Learning.
[4] Return-Based Contrastive Representation Learning for Reinforcement Learning.
6. The compared methods in the experiment are not the latest ones published.

**Questions:**

Please refer to the weakness above. I am willing to raise my score if my major concerns can be solved.

---

### Official Review · Reviewer_bVmG · 2023-11-03

**Soundness:** 3 good
**Presentation:** 1 poor
**Contribution:** 2 fair
**Rating:** 5
**Confidence:** 2

**Summary:**

This work points out the duality of visual semantic information for pixel-based RL methods.  To better utilize visual semantic information, positive unlabeled policy-guided contrast is proposed for pixel-based RL. Theoretically, authors further rethink the learning paradigm of pixel-based RL and demonstrate the proposed method can achieve a tigher lower bound of the mutual information between the policy-related information and the learned representation.

**Strengths:**

This paper delves into the affects of visual semantic informatin on the learned policy by pixel-based RL methods. By data augmentation, authors propose a positive unlabeled contrastive learning strategy and further show its theretical insights. The theoretrical analysis seems novel and insightful but not all details are easy to be understood upon my research background.

Experiments show the proposed method gains benefits over most RL environments consistently.

**Weaknesses:**

Many expressions in the article are unclear, making the content obscure and difficult to understand. To name a few, it's confusing for understanding so called "anti-redundant and policy-optimization-relevant visual semantic information" and "the information contained by the learned representation". They should be further concretized and given clear definitions accordingly.

The illustrations, such as Figrue 2, are also unclear. For figure 2, it's quite difficult to understand what differs between the top two branches and why three branches are needed?（Why can't the branch of reinforcement learning reuse the first two branches?）

**Questions:**

Besides the questions in the weakness part, one more question is that whether the proposed method is generally applicable for different RL methods? It would be better give more detailed analysis and experiments for this.

---

### Official Review · Reviewer_kGtJ · 2023-11-05

**Soundness:** 2 fair
**Presentation:** 2 fair
**Contribution:** 2 fair
**Rating:** 5
**Confidence:** 3

**Summary:**

The paper proposes a method based on contrastive learning for pixel-based reinforcement learning tasks.

**Strengths:**

The paper is easy to read and follow. The reported experimental results are convincing.

**Weaknesses:**

The citation should be given in the following sentences. “Furthermore, the encoder of DQN cannot learn the visual semantic information, and the sole reward-dependent training objective leads the encoder to learn limited semantic information from the pixel observations, resulting in the encoder training being easily trapped into a local optimum.” “So contrastive learning without negative views is more simple and effective..”

Are there any examples to prove the following sentence? “However, the intrinsic defect of the reward-based training approach is not suffciently addressed, i.e., the visual semantic information is not explored, so the training DrQ-v2 may still fall into a local optimum.”

There are triple branches in Fig. 2. Do they share the same encoder (\theta)?

Where do you reflect “anti-redundancy” in the algorithm?

I didn’t see any ablation study on data augmentation. As I understand, data augmentation sometimes matters in unsupervised learning tasks.

**Questions:**

Please see my comments in Weaknesses.